# Chronic stress in practice assistants: An analytic approach comparing four machine learning classifiers with a standard logistic regression model

Arezoo Bozorgmehr[1]*, Anika Thielmann[1,2], Birgitta Weltermann[1,2]

1 Institute of General Practice and Family Medicine, University Hospital Bonn, University of Bonn, Bonn, Germany, 2 Institute for General Medicine, University Hospital Essen, University of Duisburg Essen, Essen, Germany

* Arezoo.bozorgmehr@ukbonn.de

## Abstract

### Background

Occupational stress is associated with adverse outcomes for medical professionals and patients. In our cross-sectional study with 136 general practices, 26.4% of 550 practice assistants showed high chronic stress. As machine learning strategies offer the opportunity to improve understanding of chronic stress by exploiting complex interactions between variables, we used data from our previous study to derive the best analytic model for chronic stress: four common machine learning (ML) approaches are compared to a classical statistical procedure.

### Methods

We applied four machine learning classifiers (random forest, support vector machine, K-nearest neighbors', and artificial neural network) and logistic regression as standard approach to analyze factors contributing to chronic stress in practice assistants. Chronic stress had been measured by the standardized, self-administered TICS-SSCS questionnaire. The performance of these models was compared in terms of predictive accuracy based on the 'operating area under the curve' (AUC), sensitivity, and positive predictive value.

### Findings

Compared to the standard logistic regression model (AUC 0.636, 95% CI 0.490–0.674), all machine learning models improved prediction: random forest +20.8% (AUC 0.844, 95% CI 0.684–0.843), artificial neural network +12.4% (AUC 0.760, 95% CI 0.605–0.777), support vector machine +15.1% (AUC 0.787, 95% CI 0.634–0.802), and K-nearest neighbours +7.1% (AUC 0.707, 95% CI 0.556–0.735). As best prediction model, random forest showed a sensitivity of 99% and a positive predictive value of 79%. Using the variable frequencies at the decision nodes of the random forest model, the following five work characteristics

**Data Availability Statement:** The manuscript's data cannot be shared publicly because of ethical restrictions as our dataset includes potentially identifying information of personnel in general

practices. Data requests may be sent to the institutional ethics committee of Universitatsklinikum Bonn (ethik@ukbonn.de).

**Funding:** The authors received no specific funding for this work.

**Competing interests:** The authors have declared that no competing interests exist.

influence chronic stress: too much work, high demand to concentrate, time pressure, complicated tasks, and insufficient support by practice leaders.

## Conclusions

Regarding chronic stress prediction, machine learning classifiers, especially random forest, provided more accurate prediction compared to classical logistic regression. Interventions to reduce chronic stress in practice personnel should primarily address the identified workplace characteristics.

## 1. Introduction

Occupational stress is an important issue in health care and other workers worldwide [1]. Following stress models introduced by Selye, Lazarus and others, it was shown that chronic stress can lead to adverse (mental) health effects such as burnout or depression [2, 3]. Also, stress can produce temporary or even permanent alterations in memory [4], cognition [5], arousal/sleep [6, 7], and coping behaviours [8]. In our prior study with 214 general practitioners (GPs) and 550 practice assistants from 136 German general practices, we showed that 19.9% of the male GPs (n = 141), 35.6% of the female GPs (n = 73) and 26.4% of the practice assistants (PrAs) had high chronic stress [9]. Overall, the mean prevalence of high chronic stress was 26.3% in this workforce, which is more than twice as prevalent compared to the general population (11%) studied in the representative German Health Interview and Examination Survey for Adults (DEGS1) with more than 7.900 participants [10, 11]. Analyzing for various work and (regional) practice characteristics, we showed that only the weekly working hours correlated with high chronic stress in GPs and PrAs.

However, aiming to develop effective prevention strategies, a more profound understanding of factors causing and/or contributing to high psychological strain on an individual and group level is needed. As workplaces typically are complex and multifactorial social organizations, appropriate statistical methods are needed to analyse for complex associations and cause-effect relationships. Prior studies addressing impaired psychological well-being in primary care workers used standard statistical procedures such as prevalence ratios and logistic regression models to evaluate for associations [9, 12, 13]. These statistical approaches usually simplify the complex relationships between independent variables (features) and response variable (dependent variable): they assume that each independent variable is linked to the outcome by a linear statistical function. This is especially problematic when datasets with large numbers of non-linear interactions and interaction effects between independent variables occur, which make the model more complex [14]. Nowadays, machine learning (ML) approaches offer new opportunities to evaluate complex relationships. Conceptually, ML has the benefit that it efficiently exploits complex and non-linear interactions between variables by minimizing the error between predicted and observed response variables and improve the accuracy of the models compared to standard approaches [15, 16]. By using a large dataset available on practice assistants from our prior study, we aim to develop better understanding workplace factors, associated with chronic stress in practice assistants using machine learning. Thus, we compare four machine learning classifiers (random forest, support vector machine, K-nearest neighbors', artificial neural network) with a standard logistic regression model using standard measurements to compare test accuracy, i.e. to derive the best prediction model for chronic stress in practice assistants in primary care.

Regarding terminology, we like to point out that we use the term "prediction" as used in the context of machine learning: it refers to the output of an algorithm after it has been trained on a dataset and applied to new data to forecast the likelihood of a particular outcome. In contrast, in epidemiological analyses, a (risk) prediction model refers to a mathematical equation that uses patient characteristics (risk factors) to estimate the probability of a defined outcome prospectively.

## 2. Methods

### 2.1 Data source

The dataset used for the analyses was derived from our cross-sectional study addressing stress among general practice personnel (GPs, PrAs), which was performed among general practices belonging to the teaching practice network of the Institute for General Medicine, University Hospital Essen, Essen, Germany. A total of 764 professionals from 136 practices had taken part in the survey, which was performed in 2014. The design of the study and key results addressing the 214 GPs (practice owners and employed physicians) and 550 practice assistants (PrAs) (including medical secretaries and practice assistants in trainees) are published [9]. This analysis addresses chronic stress in 550 practice assistants (PrAs), which are the largest professional group in general practices. We documented that 26.4% of the 550 practice assistants (PrAs) had high chronic stress, as well as 19.9% of the male (n = 141) and 35.6% of the female (n = 73) general practitioners (GPs) [9]. In this workforce, the average of workers with high chronic stress was 26.3% (n = 201).

### 2.2 Ethics statement

Ethical approval for the survey had been obtained from the Ethics Committee of the Medical Faculty of the University of Duisburg-Essen (reference number: 13-5536-BO, date of approval: 24/11/2014). All participants had received written information and signed informed consent forms. The principal investigator of the study (B.W) and coauthor of this manuscript provided the data for this analysis.

### 2.3 Outcome

The primary outcome is strain due to chronic stress over the past three months. Chronic stress was measured using the German short version of the standardized, validated, self-administered TICS-SSCS questionnaire [17, 18]. This instrument measures strain due to chronic stress for the past three months. It consists of 12 items on 5-point Likert scales (0 = 'never' und 4 = 'very often'). The TICS-SSCS values are added to a sum-score. The score ranges from 0 to 48 with 0 denoting 'never stressed' and 48 'very often stressed', and reflects subjective strain due to chronic stress [17, 18]. Following the definition of chronic stress of our prior analysis, the TICS scores were dichotomized using the median (TICS = 23) as cut-off (0 = no chronic stress (TICS < 23), 1 = strain due to chronic stress (TICS ≥ 23)).

### 2.4 Socio-demographic and workplace characteristics

A total of 64 sociodemographic and workplace characteristics were used for the analyses. The sociodemographic characteristics included e.g., age, marital status, number of persons in household. Work-related characteristics comprised details on the employment (e.g., number of hours per week, work status, employment contract), duties in practice (e.g., reception, telephone, prescription, blood pressure measurement) and subjective perceptions of workload (e.g., self-determination of sequence of work steps, influence on work assigned, plan the work

independently). The standardized 'short questionnaire for workplace analysis' (German: Kurz-fragebogen zur Arbeitsanalyse (KFZA)) was used to assess workplace characteristic [19]. For details on the work characteristics see Tables 1–3. In line with the TICS instrument, which addresses strain due to chronic stress during the past three months, all workplace characteristics had been requested regarding the past three months (see Table 4).

## 2.5 Statistical analysis

**2.5.1 Handling of missing data.** Missing values were observed in 0.2% to 11%. If missing data were above 5%, this is indicated in the Tables 1–3. Common imputation methods for supervised learning were applied to handle missing data [20]. The K-nearest neighbors algorithm was used for imputing missing values in TICS scores with k = 10. For continuous variables we used median imputation and for categorical variables a separate category 'unknown' [20].

**2.5.2 Preparation of datasets for machine learning.** After pre-processing the data to compare machine learning classifiers, the dataset was split into a 'training' and a 'validation' dataset. Fig 1 illustrates the study process flow. We used the 10-fold cross validation approach

**Table 1. Sociodemographic characteristics of practice assistants (n = 550) and strain due to chronic stress (measured by the standardized and validated TICS tool): Items and sum scores.**

| | Participants (N = 550) | | |
|---|---|---|---|
| *Continuous variables* | **Mean** | **SD** | **Range** |
| Age | 38 | 12.61 | 16–71 |
| Persons in household more age 18 | 2 | 1.12 | 0–6 |
| Persons in household below age 18 | 1 | 0.84 | 0–6 |
| Number of physicians in practice | 3 | 2.16 | 1–10 |
| Number of practice assistants in practice | 8 | 7.66 | 0–35 |
| *Categorical variables* | **n** | | **%** |
| **Female gender** | 544 | | 99.3 |
| **Marital status** | | | |
| Married | 277 | | 50.4 |
| Single | 221 | | 40.2 |
| Divorced | 45 | | 8.2 |
| Widowed | 7 | | 1.3 |
| **Number of persons in household** | 72 | | 13.1 |
| **Cares for next of kin** | 75 | | 13.6 |
| **Working hours/week** | | | |
| 1–9 hours | 12 | | 2.2 |
| 10–19 hours | 52 | | 9.5 |
| 20–29 hours | 116 | | 21.1 |
| 30–39 hours | 221 | | 40.2 |
| 40–49 hours | 116 | | 21.1 |
| 50–59 hours | 12 | | 2.2 |
| >60 hours | 10 | | 1.8 |
| **Working full time** | 364 | | 66.2 |
| **Has open-ended employment contract** | 466 | | 84.7 |
| **Had participated in stress seminar in the past** | 31 | | 5.6 |
| **Had used counseling for stress reduction** | 50 | | 9.1 |
| **High strain due to chronic stress (TICS $\geq$ 23)** | 125 | | 22.7 |

**Table 2. Practice and workplace characteristics during the past three months (n = 550 practice assistants).**

| Practice characteristics | | |
|---|---|---|
| **Practice structure** | | |
| Working in group practice | 296 | 53.8 |
| Working in single physician practice | 147 | 26.7 |
| Working in practice with several locations | 50 | 9.1 |
| Working in practice with an employed physician | 39 | 7.1 |
| Working in privately owned health center | 6 | 1.1 |
| **Medical records** | | |
| Electronic medical records (EHR) | 348 | 63.3 |
| Paper and electronic records | 187 | 34.0 |
| **Practice services** | | |
| Emergent home visits | 515 | 93.6 |
| Practice offers regular home visits | 511 | 92.9 |
| Nursing home visits* | 508 | 92.4 |
| **Tasks of practice assistant during past 3 months** | | |
| Scheduled appointments | 518 | 94.2 |
| Documented in patients´ EHR | 513 | 93.3 |
| Prepared prescriptions | 504 | 91.6 |
| Pulled up paperhealth records or opened electronic patient files | 500 | 90.9 |
| Performed phone service | 499 | 90.7 |
| Worked at reception | 486 | 88.4 |
| Obtained blood pressure readings | 461 | 83.8 |
| Performed ECGs | 430 | 78.2 |
| Prepared practice equipment for the day and switch them off in the evening | 414 | 75.3 |
| Performed laboratory work | 393 | 71.5 |
| Supported physician during patient-consultations | 363 | 66.0 |
| Supported billing of statutory health insurance patients | 358 | 65.1 |
| Performed disease-management examinations | 332 | 60.4 |
| Applied long-term blood pressure devices* | 327 | 59.5 |
| Ordered medical supply | 284 | 51.6 |
| Applied long-term ECG* | 247 | 44.9 |
| Ordered office supply | 239 | 43.5 |
| Performed treadmill testing | 237 | 43.1 |
| Supported billing of private patients* | 236 | 42.9 |
| Performed doppler examination of foot vessels/measured ankle-arm index* | 103 | 18.7 |

*Missing values above 5%

in machine learning models to measure the unbiased prediction accuracy of the models (see Fig 2). Based on the literature, 10 was chosen as optimal number of folds, which optimizes the time to complete the test while minimizing the bias and variance associated with the validation process [21–23]. The K-Fold cross validation method also called rotation estimation is used to minimize the bias associated with the random sampling of the training and holdout data samples in comparing the predictive accuracy of two or more machine learning methods. In this method the complete dataset (D) is randomly split into k mutually exclusive subsets (the folds: D1, D2,. . ., Dk) of approximately equal size. The classification model is trained and tested k times. Each time (t 2 {1, 2,. . ., k}), it is trained on all but one folds (Dt) and tested on the

**Table 3. Self-assessment of workplace situation (n = 550 practice assistants).**

| Work aspects | Workplace factor | Mean Score (PrAs) | 95% CI |
|---|---|---|---|
| Job content | Versatility | 3.6 | 3.58–3.7 |
| | Completeness of task | 3.5 | 3.41–3.57 |
| Resources | Scope of action | 3.4 | 3.37–3.49 |
| | Social support | 4.0 | 3.98–4.12 |
| | Cooperation | 3.6 | 3.53–3.66 |
| Stressors | Qualitative work demands | 2.2 | 2.14–2.29 |
| | Quantitative work demands | 2.9 | 2.83–3.01 |
| | Work disruptions | 2.7 | 2.67–2.81 |
| | Workplace environment | 2.2 | 2.13–2.3 |
| Organizational culture | Information and participation | 3.6 | 3.57–3.73 |
| | Benefits | 2.9* | 2.77–2.94 |

remaining single fold (Dt). The cross validation estimate of the overall accuracy is calculated as the average of the k individual accuracy measures by formula:

$$CVA = \sum_{i=1}^{k} A_i \qquad (1)$$

Where CVA stands for cross-validation accuracy, k is the number of folds used, and A is the accuracy measure of each fold [21].

**2.5.3 Logistic regression as standard statistical procedure.** *Logistic Regression (LR)* is a classical statistical modelling procedure to analyze one dependent dichotomous or binary outcome and one or more nominal, ordinal, interval or ratio-level independent variables. LR models are frequently applied to exposure-event studies in medical research, because they can be used to estimate the model predictors' odds ratio [24]. All variables significant in bivariate analysis were included in the logistic regression model.

**2.5.4 Machine learning approaches.** *1) K-Nearest Neighbors (KNN)* classifies an object by a majority vote of its neighbors, with the object being assigned to the class most common amongst its k nearest neighbors (k is a positive integer). If k = 1, the object is simply assigned to the class of its nearest neighbor. KNN is a type of instance-based or lazy learning where the

**Table 4. Chronic stress of practice assistants: Results of TICS (Trierer Inventory of Chronic Stress) (n = 550).**

| How often in the last 3 months did you experience . . . | Never | Rarely | Sometimes | Frequently | Very Frequently |
|---|---|---|---|---|---|
| | n(%) | n(%) | n(%) | n(%) | n(%) |
| Fear, something unpleasant might occur | 72 (13.1) | 213 (38.7) | 190 (34.5) | 54 (9.8) | 21 (3.8) |
| Lack of recognition for good performance | 158 (28.7) | 157 (28.5) | 121 (22.0) | 71 (12.9) | 42 (7.6) |
| Times with too many obligations | 38 (6.9) | 119 (21.6) | 167 (30.4) | 157 (28.5) | 67 (12.2) |
| Times when being unable to suppress worrying thoughts | 90 (16.4) | 174 (31.6) | 182 (33.1) | 83 (15.1) | 21 (3.8) |
| Work is not appreciated despite doing the best | 157 (28.5) | 200 (36.4) | 116 (21.1) | 56 (10.2) | 20 (3.6) |
| Everything is too much | 86 (15.7) | 174 (31.7) | 174 (31.7) | 85 (15.5) | 30 (5.5) |
| Times of worry and one cannot stop it | 138 (25.1) | 186 (33.9) | 139 (25.3) | 57 (10.4) | 29 (5.3) |
| Times when being unable to perform as expected | 120 (21.8) | 299 (54.4) | 107 (19.5) | 19 (3.5) | 5 (0.9) |
| Times in which the responsibility for others is a burden | 162 (29.5) | 215 (39.1) | 123 (22.4) | 42 (7.6) | 8 (1.5) |
| Times when the work gets too much | 85 (15.5) | 205 (37.3) | 183 (33.3) | 60 (10.9) | 17 (3.1) |
| Fear of not being able to perform the tasks | 126 (22.9) | 229 (41.6) | 137 (24.9) | 43 (7.8) | 15 (2.7) |
| Times when being overwhelmed with worries | 165 (30.0) | 189 (34.4) | 128 (23.3) | 45 (8.2) | 23 (4.2) |

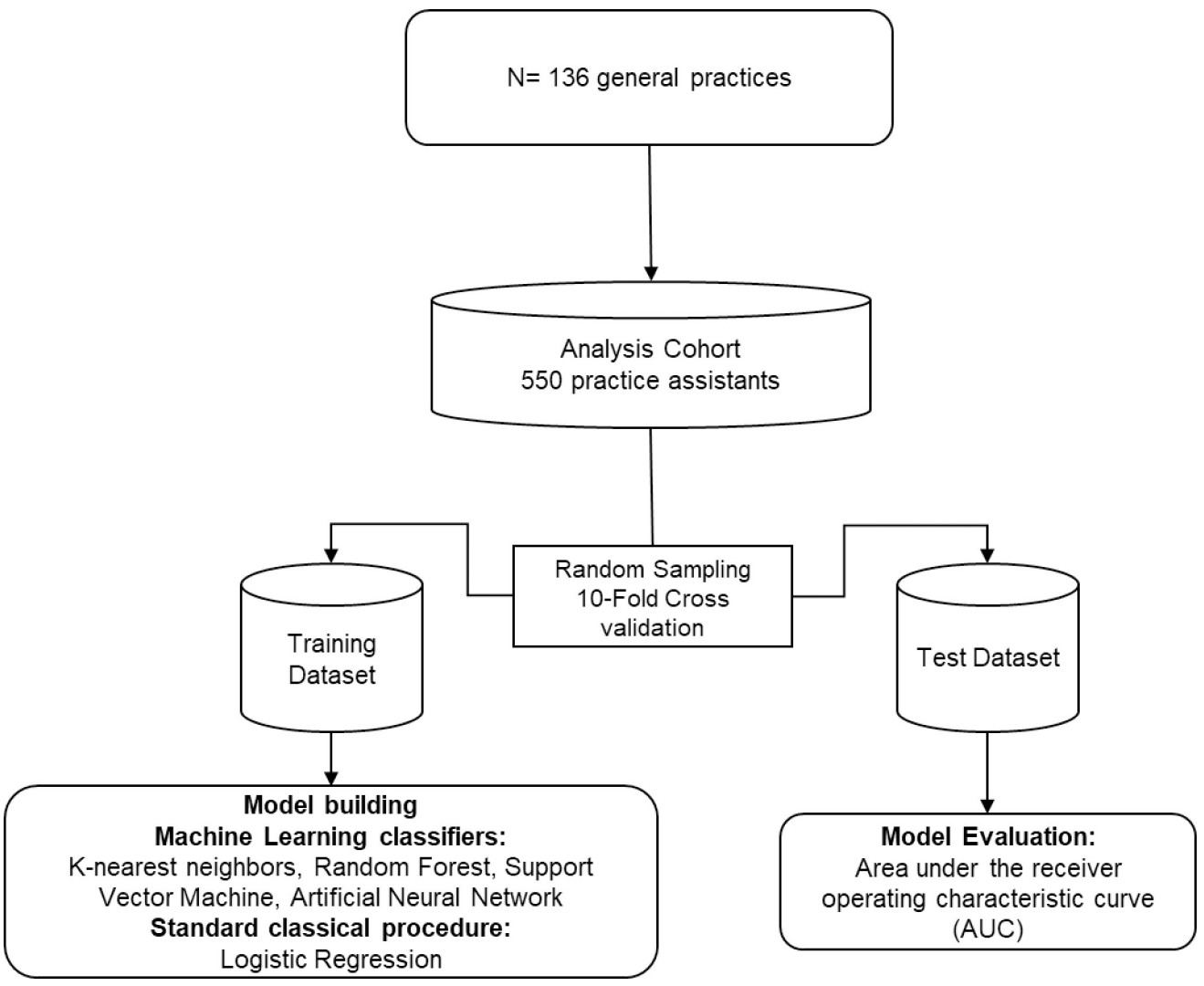

**Fig 1. Machine learning data extraction process flow.**

function is only approximated locally and all computation is deferred until classification [25, 26]. In this study, we used KNN applying k = 10 neighbors, which are the ten closest observations in multidimensional space based on Euclidean distance function to model the training dataset.

*2) Support Vector Machine (SVM)* represents different outcome classes in a hyperplane in multidimensional space to find the maximum marginal hyperplane. SVM generates the hyperplane in an iterative manner to minimize the error. A basic SVM is a non-parametric linear classifier that creates a hyperplane using the Euclidean distance function from the nearest input values to determine the target states. In order to obtain probability estimates, a logistic regression model is fitted to the output of the support vector machine [25]. In this study, the SVM classifier used RBF (Radial basis function) kernel, a training error of 1.0E-12, and a default boundary tolerance of a 1.0E-03 hyperplane. To obtain proper probability estimates, we used the option that fits calibration models to the outputs of the SVM.

*3) Random Forest (RF)* is a collection of decision trees, each constructed in a bootstrapped sample and from a random subset of the possible predictors at each node. RF is used to reduce

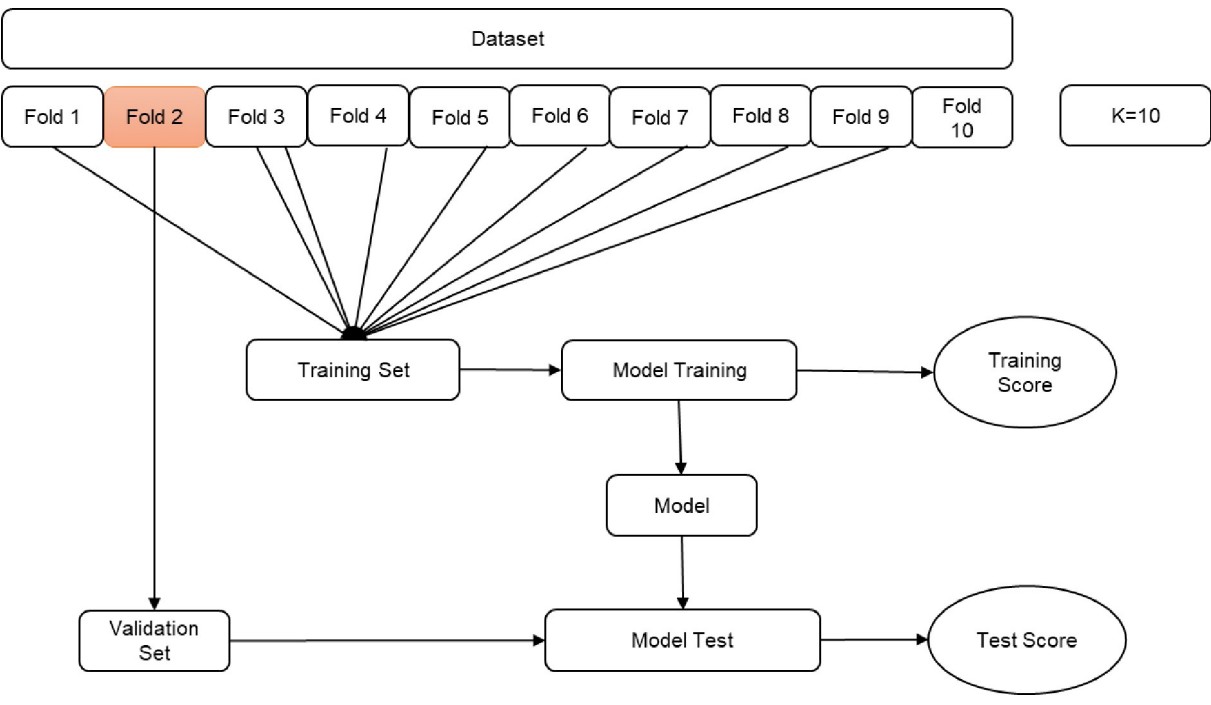

**Fig 2. K-Fold cross validation.**

variance associated with decision trees [27, 28]. In this study, the forest is constructed consisting of randomly 1,000 individual trees. A large number of trees increases the predictive accuracy of RF models and the forest does not require extensive tuning [29]. Due to the insensitivity of error rates to the number of features selected to split each node, we used the default of a random sample of $\sqrt{n}$ of predictors at each node with n being the total number of predictors under consideration. The predicted probability was derived based on average prediction across all of the trees.

*4) Artificial Neural Network (ANN)* is a computational and flexible model that expresses complex non-linear relationships among features, which consist of an interconnected group of variables. A basic ANN model consists of three layers of neurons, i.e. input, output, and hidden layer. These layers can learn from data iteratively through a backpropagation classifier. It trains a multilayer perceptron with one hidden layer, an input layer with the number of nodes equal to the sum of features, and an output layer [30]. This study used a multilayer Perceptron classifier with one hidden layer, a learning rate value with decay of 0.3, and a momentum rate for the backpropagation classifier of 0.2. Suitable ranges for these parameters are within 0.15–0.8 for learning rate and 0.1–0.4 for momentum [30].

Development of the models was completed using Python (Version 3.7.3) and Python's Scikit-Learn library (https://scikit-learn.org/stable/).

## 3. Results

### 3.1 Sociodemographic and workplace characteristics of the study population

The dataset comprised results of 550 PrA from 136 general practices. The vast majority of the total of PrAs were females (98.9%) with a mean age of 38 years (SD 12.6). Regarding the

marital status, 50.6% (n = 277) of the PrAs were married. On average, they worked in the current practice for 18.8 years (SD 12.5), 32.5% in part-time.

## 3.2. Primary outcome: Strain due to chronic stress

The TICS score of the population ranged from 0 to 44 with a mean of 17.2 and median of 17.0. In the total dataset, 22.7% (n = 125) had high strain due to chronic stress versus 77.3% (n = 425) low strain due to chronic stress. Regarding socio-demographic characteristics personnel with high strain due to chronic stress showed the following significant differences compared to those with low strain: older PrAs (mean 38.76) vs. younger PrAs (mean 24.36), unmarried PrAs (29.4%) vs. married PrAs (17%). While caring for next of kin did not differ between groups. No gender-specific distribution was applied, because PrAs were predominantly female (98.9%). All regression and machine learning approaches were applied to the dataset with female subjects only (n = 546).

## 3.3. Results of four machine learning classifiers

**3.3.1 Prediction accuracy.**   The performance of the machine learning classifiers was assessed using the validation dataset by calculating Harrell's c-statistic, a measure of the total area under the receiver operating characteristic curve (AUC) [31]. The results showed an AUC of 0.844 (95%CI, 0.684–0.843) for RF, 0.760 (95%CI, 0.605–0.777) for ANN, 0.787 (95%CI, 0.634–0.802) for SVM, and 0.707 (95%CI, 0.556–0.735) for KNN.

**3.3.2 Classification analysis.**   Corresponding results of sensitivity and positive prediction value (PPV) for machine learning were 99% and 79% for RF, 87% and 85% for ANN, 87% and 86% for the SVM, and 99% and 78% for KNN.

## 3.4. Results of Logistic regression analysis

In bivariate analysis, the following factors were associated with strain significantly: persons in household below age 18, marital status, age, working hours/week, room equipment, work status, performed laboratory work, obtained blood pressure readings, and performed doppler examination of foot vessels/measured ankle-arm index as duties in practice. C statistics for logistic regression showed an AUC of 0.636 (95%CI, 0.490–0.674). This model predicted 316 cases correctly from 425 total cases, with a sensitivity of 75% and positive prediction value (PPV) of 44%.

## 3.5. Comparison of ML and regression analysis

The prediction accuracy according to the discrimination (AUC c-statistic) value is shown in Table 5 for all models. All machine learning models achieved statistically improvements in compared to the standard logistic regression model: +20.8% for RF, +15.1% for SVM, +12.4% for ANN, and +7.1% for KNN. Random forest is performing well out of all four machine learning classifiers. RF classifier resulted in a net increase of 104 strain due to chronic stress cases from the logistic regression baseline model, increasing the sensitivity to 99% and PPV to 79%. See Table 6 for more details of machine learning models.

## 3.6. Variable rankings in machine learning models

Of the 4 ML approaches used, variable importance can only be determined in artificial neural network and random forest. Artificial neural network model uses the overall weighting of the variables within the model. Random forest ranks variable importance based on decision-trees on the selection frequency of the variable as a decision node. For KNN does not provide a

**Table 5. Performance of the machine learning algorithms predicting chronic stress derived from applying training algorithms on the validation dataset.** Higher c-statistics results in better algorithm discrimination. The baseline (BL) standard logistic regression model is provided for comparative purposes.

| Algorithms | AUC c-statistic | 95% Confidence Interval | | Absolute change in AUC (%) |
|---|---|---|---|---|
| | | LCL | UCL | |
| BL: Logistic Regression | 0.636 | 0.490 | 0.674 | [Reference] |
| ML: K-nearest Neighbours | 0.707 | 0.556 | 0.735 | +7.1% |
| ML: Support Vector Machine | 0.787 | 0.634 | 0.802 | +15.1% |
| ML: Artificial Neural Network | 0.760 | 0.605 | 0.777 | +12.4% |
| ML: Random Forest | 0.844 | 0.684 | 0.843 | +20.8% |

method for the importance or coefficients of variables. We used a nonlinear SVM classifier with RBF kernel, which has no variable importance methods. The variable importance was determined by the coefficient effect size for logistic regression model. The identified factors such as persons in household below age 18, age below 35 years old, and insufficient room equipment that have identified by logistic regression, has also identified by ANN and RF. The most determined factors by both of ANN and RF included work related characteristics such as too much work, high demand to concentrate, time pressure, complicated tasks, and insufficient practice room conditions (See Table 7).

## 4. Discussion

To the best of our knowledge, this study is the first to use machine learning for a better understanding of stress in primary care practice personnel. Comparing four common machine learning (ML) approaches to a classical statistical procedure, we showed that all four machine learning approaches provided more accurate models for the prediction of strain due to chronic stress than as standard regression analysis. Random forest showed the highest accuracy with workload, high demand to concentrate, and time pressure being the most important factors associated with chronic stress. These factors were also identified in other studies in the target populations GPs and GP practice personnel. Addressing job satisfaction, Harris et al. identified time pressure as the most frequent stressor in a study with 626 Australian practice staff in 96 general practices [12]. Studying 158 Canadian family physicians, Lee et al. determined the following occupational stressors as relevant: challenging patients, high workload, time limitations, competency issues, challenges of documentation and practice management and changing roles within the workplace [13, 32]. Similarly, Hoffmann et al. showed that the work disruption was a negative relevant workplace factor in study with 550 practice assistants [33].

**Table 6. Full details on classification analysis.**

| Algorithms | Chronic stress cases correct (True Positive) | Chronic stress cases incorrect (False Negative) | Total chronic stress cases | Non-chronic stress cases correct (True Negative) | Non-chronic stress cases incorrect (False Positive) | Total non-chronic stress cases | Sensitivity (True Positive) | Positive Predictive Value (PPV) |
|---|---|---|---|---|---|---|---|---|
| Logistic Regression | 316 | 109 | 425 | 68 | 57 | 125 | 0.751 | 0.440 |
| ML: Random Forest | 420 | 5 | 425 | 15 | 110 | 125 | 0.988 | 0.792 |
| ML: K-nearest Neighbours | 421 | 4 | 425 | 6 | 119 | 125 | 0.991 | 0.780 |
| ML: Support Vector Machine | 369 | 56 | 425 | 66 | 59 | 125 | 0.868 | 0.862 |
| ML: Artificial Neural Network | 369 | 56 | 425 | 59 | 66 | 125 | 0.868 | 0.848 |

**Table 7. The most influential predictor variables associated with chronic stress listed by coefficient effect size (Standard logistic regression) weighting (Artificial neural network) and selection frequency (Random forest).**

| Standard model | | Machine learning models | | | |
|---|---|---|---|---|---|
| Logistic regression | Coefficient | Artificial Neutral Network | Weight (%) | Random Forest | Frequency |
| Obtained blood pressure readings | 0.951 | Too much work | 39.7 | Too much work | 0.73 |
| Persons in household below age 18 | 0.349 | High demand to concentrate | 39.3 | High demands to concentrate | 0.71 |
| Working hours/week more than 40 | 0.121 | Time pressure | 36.7 | Time pressure | 0.70 |
| Work status | -0.109 | Complicated tasks | 31.5 | Complicated tasks | 0.67 |
| Performed laboratory work | 0.091 | Insufficient practice room conditions | 18.1 | Age ≤ 35 | 0.63 |
| Employment contract | 0.063 | Interrupted during work | 14.9 | Insufficient support by practice leaders | 0.52 |
| Age ≤ 35 | 0.045 | Persons in household below age 18 | 13.8 | Insufficient workplace environment | 0.51 |
| Insufficient workplace environment | 0.028 | Working hours/week more than 40 hours | 12.7 | Insufficient practice room conditions | 0.50 |
| Performed doppler examination of foot vessels/measured ankle-arm index | 0.018 | Workplace environment | 12.3 | Holding together well | 0.48 |
| Marital status/single | 0.006 | Number of practitioners in the practice | 10.6 | Influence on work assigned | 0.43 |

These stressors are described to influence poor physician well-being and adverse patient outcomes such as low patient satisfaction [34]. The relevance of such chronic psychological burden is tremendous as it was shown that physiological responses due to stress negatively affect e.g. memory, immune system functions, the function of the cardiovascular system, and brain electric activity [35, 36].

## 4.1 Comparison to other ML analyses

There are a few other studies from other medical fields, which compared standard statistical and ML approaches, similar to our results. Machine learning is considered a branch of artificial intelligence, which extracts meaningful patterns from data and develops prediction models using several algorithms [37]. ML approaches integrate many different levels of data to develop a new approach to classification based on medical issues such as chronic stress and linked more precisely to interventions for a given individual. Better model accuracy by machine learning was also found in an UK study on cardiovascular risk prediction. Using routine clinical data of 378,256 patients four machine learning algorithms (random forest, logistic regression, gradient boosting, and neural network) were compared to an established algorithm (American College of Cardiology guidelines) to predict first cardiovascular event over 10-years [38]. Neural network performed best, with a predictive accuracy improving by 3.6% compared to baseline algorithm. Using a dataset with 9.502 heart failure patients and a one-year follow-up, a US study compared four machine learning methods (least absolute shrinkage and selection operation regression, classification and regression trees, random forests, and gradient boosted modeling (GBM)) with logistic regression as a classical statistical procedure to predict four heart failure outcomes. The C statistic results for all outcomes show that ML methods were better calibrated and that gradient-boosted (GMB) model was the most consistent ML modeling approach [39]. In the field of oncology, a large American study on breast cancer survival compared two ML algorithms (artificial neural network and decision trees) to classical statistical logistic regression using a large dataset with more than 200,000 cases. The decision tree approach was the best predictor with 93.6% prediction accuracy, followed by

artificial neural network with 91.2% and LR with 89.2% [40]. Overall, machine learning approaches yielded more accurate results than classical methods in our and the above-mentioned studies.

## 4.2 Strength and limitations

The key strength of this study is the comparison of a range of machine learning approaches in the field of healthcare workers´ well-being. Chronic stress measurement approaches based on self-reported questionnaires [17, 41] are subjective and cannot provide immediate information about the state of a person. A continuous stress monitoring using data mining technology helps to better understand stress patterns and also provide better insights about possible future interventions.

Limitations of this study include the rather small sample size and the large number of predictor variables (features), which poses a risk for overfitting [42, 43]. One of the key components of predictive accuracy is the amount and quality of the data to provide better results. Furthermore, our data source contained practice assistants from the German region only, which limits generalizability and requires validation in populations from other countries where job tasks and challenges might be different. Although the data collection was conducted in 2014, the results still apply to German practices, except that the COVID pandemic likely increased workload and psychological burden, which we are currently evaluating in an ongoing study [11]. Prospectively, research using continuous stress monitoring and data mining technologies will help to better understand stress patterns and provide even deeper insights for possible future interventions.

## 5. Conclusion

Compared to logistic regression as a classical statistical procedure, this study showed that all machine learning classifiers provided more accurate models for the prediction of chronic stress in practice assistants with random forest performing best. Identification of chronic stress is of importance for the well-being and productivity of practice assistants. RF identified prominent predictor variables (features) that influence chronic stress which should be considered when developing interventions to reduce chronic stress.

## Acknowledgments

We would like to thank all participating practices for their support of the study.

## Author Contributions

**Conceptualization:** Arezoo Bozorgmehr.

**Data curation:** Arezoo Bozorgmehr.

**Formal analysis:** Arezoo Bozorgmehr.

**Methodology:** Arezoo Bozorgmehr, Birgitta Weltermann.

**Project administration:** Birgitta Weltermann.

**Software:** Arezoo Bozorgmehr.

**Supervision:** Birgitta Weltermann.

**Validation:** Arezoo Bozorgmehr.

**Visualization:** Arezoo Bozorgmehr.

**Writing – original draft:** Arezoo Bozorgmehr.

**Writing – review & editing:** Anika Thielmann.

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
