## [Decision Letter · Decision Letter 0]

23 Dec 2020

PONE-D-20-23593

Chronic stress in practice assistants: an Analytic approach comparing four machine learning classifiers with a standard logistic regression model

PLOS ONE

Dear Dr. Bozorgmehr,

Thank you for submitting your manuscript to PLOS ONE. After careful consideration, we feel that it has merit but does not fully meet PLOS ONE’s publication criteria as it currently stands. Therefore, we invite you to submit a revised version of the manuscript that addresses the points raised during the review process.

We look forward to receiving your revised manuscript.

Kind regards,

Alfredo Vellido

Academic Editor

PLOS ONE

Journal Requirements:

2. Please clarify in your Methods section how the dataset was obtained by the authors of this study, and whether there was any ethical oversight over the data collection for this study. Please state whether or not the authors had access to any identifying information.

3.We note that you have indicated that data from this study are available upon request. PLOS only allows data to be available upon request if there are legal or ethical restrictions on sharing data publicly. For information on unacceptable data access restrictions, please see http://journals.plos.org/plosone/s/data-availability#loc-unacceptable-data-access-restrictions.

Reviewers' comments:

Reviewer's Responses to Questions

**Comments to the Author**

1. Is the manuscript technically sound, and do the data support the conclusions?

Reviewer #1: Yes

Reviewer #2: Yes

2. Has the statistical analysis been performed appropriately and rigorously? 

Reviewer #1: Yes

Reviewer #2: Yes

3. Have the authors made all data underlying the findings in their manuscript fully available?

Reviewer #1: Yes

Reviewer #2: Yes

4. Is the manuscript presented in an intelligible fashion and written in standard English?

Reviewer #1: Yes

Reviewer #2: Yes

5. Review Comments to the Author

Reviewer #1: Dear Authors

the manuscript entitled"Chronic stress in practice assistants: an Analytic approach comparing four machine

learning classifiers with a standard logistic regression model" has several interesting and strategic findings and i certainly used it in the real life. the chronic stress is very important issue in our society for health. thank you for your study.

some minor revision is only needed. although the manuscript is too long and the part of method was explained in the details but i do not recommend to summarize because it help some readers who not know intelligent artificial method.

another limitation that you mentioned is that certainly the important features of work in the chronic stress is dependent to several factor of social-psychological-economy- culture. then these factors were important in that community or culture. however the results can give us a closer look at reality. i suggest that the researchers analyse again the classifiers only in the women subjects and clear the men. i think the accuracy of results increases.

Abstract: the background of abstract should be brief and general and the explanation of study add to the method. the names of questionnaires enter in the method.

introduction: change the phrase of chronic strain to chronic stress. strain is force that could cause stress and stress is mental state and used as impairment.

results: the table 6: the stress is add to this topic:Total Non-(stress) Cases =

if possible add the weight to main features listed in the table 7

discussion: as a strategic note you add the some complications due to chronic stress for general health or some studies that measure the effect of stress on brain function or other parts

The impact of stress on body function: A review

H Yaribeygi, Y Panahi, H Sahraei, TP Johnston, A Sahebkar

EXCLI journal 16, 1057

Support vector machine classification of brain states exposed to social stress test using EEG-based brain network measures

S Lotfan, S Shahyad, R Khosrowabadi, A Mohammadi, B Hatef

Biocybernetics and Biomedical Engineering 39 (1), 199-213

Reviewer #2: Dear authors, your manuscript is interesting but I need you to answer some questions:

INTRODUCTION

- The introduction is very short. The constructs and concepts necessary to understand the manuscript are not explained.

- Page 4, paragraph 1, lines 47-50: this information should go in the “Method” section.

METHODS

Data source:

- What categories are there among the "general practice personnel"? The authors must describe the sample used.

- What was the target population? How was the sample chosen? The authors must specify it.

DISCUSSION

Page 21, lines 272-297: The first two paragraphs do not contribute anything new and repeat information about the results.

Limitations

The information is from 2014. There was a global economic crisis that affected working conditions. It should be said since currently, working conditions are not equivalent.

REFERENCES

Many bibliographies are obsolete and some citations are incomplete. The bibliographic citations used are more than 5 years old (57,1%). The authors must update and arrange the bibliography.

Too many references do not meet the journal guidelines and that have errors. The authors should review this section.

6. PLOS authors have the option to publish the peer review history of their article (what does this mean?). If published, this will include your full peer review and any attached files.

Reviewer #1: **Yes: **Boshra Hatef

Reviewer #2: No

---

## [Author Response · Author response to Decision Letter 0]

16 Feb 2021

Response to Editor

PONE-D-20-23593 

Chronic stress in practice assistants: an Analytic approach comparing four machine learning classifiers with a standard logistic regression model

PLOS ONE

Dear Dr. Vellido, 

We like to thank you and the reviewers for the very helpful suggestions. 

Please find enclosed our revision and answers to the open items.

Best regards, 

Arezoo Bozorgmehr

Editor comments:

1. If applicable, we recommend that you deposit your laboratory protocols in protocols.io to enhance the reproducibility of your results. Protocols.io assigns your protocol its own identifier (DOI) so that it can be cited independently in the future.

Reply: Not applicable.

Reply: Done. 

3. Please clarify in your Methods section how the dataset was obtained by the authors of this study, and whether there was any ethical oversight over the data collection for this study. Please state whether or not the authors had access to any identifying information.

Reply: Thank you, we clarified this. The information regarding to the ethics statement was already included (please see P.7, lines 130-134). We now highlighted this by inserting in a new headline and added requested aspects. 

Text: P. 7, lines 131-136 (document of revised Manuscript with track changes):

“2.2 Ethics statement: 

Ethical approval for survey had been obtained from the Ethics Committee of the Medical Faculty of the University of Duisburg-Essen (reference number: 13-5536-BO, date of approval: 24/11/2014). All participants had received written information and signed informed consent forms. The principal investigator of the study (B.W) and coauthor of this manuscript provided the data for this analysis.”

Reply: The manuscript’s data cannot be shared publicly because of ethical restrictions as our dataset includes potentially identifying information of personnel in general practices. Data requests may be sent to the institutional ethics committee (ethik@ukbonn.de).

a. If there are ethical or legal restrictions on sharing a de-identified data set, please explain them in detail (e.g., data contain potentially identifying or sensitive patient information) and who has imposed them (e.g., an ethics committee). Please also provide contact information for a data access committee, ethics committee, or other institutional body to which data requests may be sent.

Reply: The manuscript’s data cannot be shared publicly because of ethical restrictions as our dataset includes potentially identifying information of personnel in general practices. Data requests may be sent to the institutional ethics committee (ethik@ukbonn.de).

Reviewer reports: 

Reviewer #1:

1. although the manuscript is too long and the part of method was explained in the details but i do not recommend to summarize because it help some readers who not know intelligent artificial method.

Reply: Thank you.

2. another limitation that you mentioned is that certainly the important features of work in the chronic stress is dependent to several factor of social-psychological-economy- culture. then these factors were important in that community or culture. however the results can give us a closer look at reality. i suggest that the researchers analyse again the classifiers only in the women subjects and clear the men. i think the accuracy of results increases.

Reply: Thank you for this suggestion. We fully agree and had used only the dataset with the women for our analyses. This is outlined in the manuscript. 

Revised text: Please see section 3.2, line 254-257 (document of revised Manuscript with track changes).

“No gender-specific distribution was applied, because PrAs were predominantly female (98.9%). All regression and machine learning approaches were applied to the dataset with female subjects only (n=546).”

3. Abstract: the background of abstract should be brief and general and the explanation of study add to the method. the names of questionnaires enter in the method.

Reply: Thank you for your suggestion, we changed this. 

Revised text: Please see abstract (document of revised Manuscript with track changes):

 “Background

 Occupational stress is associated with adverse outcomes for medical professionals and patients. In our cross-sectional study with 136 general practices, 26.4% of 550 practice assistants showed high chronic stress. As machine learning strategies offer the opportunity to improve understanding of chronic stress by exploiting complex interactions between variables, we used data from our previous study to derive the best analytic model for chronic stress: four common machine learning (ML) approaches are compared to a classical statistical procedure.”

 “Methods

We applied four machine learning classifiers (random forest, support vector machine, K-nearest neighbors’, and artificial neural network) and logistic regression as standard approach to analyze factors contributing to chronic stress in practice assistants. Chronic stress had been measured by the standardized, self-administered TICS-SSCS questionnaire. The performance of these models was compared in terms of predictive accuracy based on the ‘operating area under the curve’ (AUC), sensitivity, and positive predictive value.”

4. Introduction: change the phrase of chronic strain to chronic stress. strain is force that could cause stress and stress is mental state and used as impairment.

Reply: Done

Revised text: Please see page 5, line 104 (document of revised Manuscript with track changes): “… associated with chronic stress in practice assistants …”

5. results: the table 6: the stress is add to this topic:Total Non-(stress) Cases =

Reply: Thank you for your hint, we corrected this.

Revised text: Please see table 6.

6. if possible add the weight to main features listed in the table 7

Reply: We added this information as suggested, thank you.

Revised text: Please see table 7.

7. discussion: as a strategic note you add the some complications due to chronic stress for general health or some studies that measure the effect of stress on brain function or other parts

The impact of stress on body function: A review

H Yaribeygi, Y Panahi, H Sahraei, TP Johnston, A Sahebkar

EXCLI journal 16, 1057

Support vector machine classification of brain states exposed to social stress test using EEG-based brain network measures

S Lotfan, S Shahyad, R Khosrowabadi, A Mohammadi, B Hatef

Biocybernetics and Biomedical Engineering 39 (1), 199-213

Reply: Thank you for these interesting articles, which we added to our paper.

Revised text: Please see references on page 22, lines 352-354 (document of revised Manuscript with track changes).

“The relevance of such chronic psychological burden is tremendous as it was shown that physiological responses due to stress negatively affect e.g. memory, immune system functions, the function of the cardiovascular system, and brain electric activity [35,36].”

Reviewer #2: 

8. INTRODUCTION

a) The introduction is very short. The constructs and concepts necessary to understand the manuscript are not explained.

Reply: Thank you, we revised the text profoundly. First, we refer to the construct of stress as developed by Selye and Lazarus. Second, we outline the construct of practices being multi-parameter systems, which affect professionals working there. Third, we outlined the concept of machine learning as analytic strategy more in detail.

Revised text: Please see pages 4-6, lines 59-115 (document of revised Manuscript with track changes): 

“Occupational stress is an important issue in health care and other workers worldwide [1]. Following stress models introduced by Selye, Lazarus and others, it was shown that chronic stress can lead to adverse (mental) health effects such as burnout or depression [2,3]. Also, stress can produce temporary or even permanent alterations in memory [4], cognition [5], arousal/sleep [6,7], and coping behaviours [8]. In our prior study with 214 general practitioners (GPs) and 550 practice assistants from 136 German general practices, we showed that 19.9% of the male GPs (n = 141), 35.6% of the female GPs (n = 73) and 26.4% of the practice assistants (PrAs) had high chronic stress [9]. Overall, the mean prevalence of high chronic stress was 26.3% in this workforce, which is more than twice as prevalent compared to the general population (11%) studied in the representative German Health Interview and Examination Survey for Adults (DEGS1) with more than 7.900 participants [10,11]. Analyzing for various work and (regional) practice characteristics, we showed that only the weekly working hours correlated with high chronic stress in GPs and PrAs. 

However, aiming to develop effective prevention strategies, a more profound understanding of factors causing and/or contributing to high psychological strain on an individual and group level is needed. As workplaces typically are complex and multifactorial social organizations, appropriate statistical methods are needed to analyse for complex associations and cause-effect relationships. Prior studies addressing impaired psychological well-being in primary care workers used standard statistical procedures such as prevalence ratios and logistic regression models to evaluate for associations [9,12,13]. These statistical approaches usually simplify the complex relationships between independent variables (features) and response variable (dependent variable): they assume that each independent variable is linked to the outcome by a linear statistical function. This is especially problematic when datasets with large numbers of non-linear interactions and interaction effects between independent variables occur, which make the model more complex [14]. Nowadays, machine learning (ML) approaches offer new opportunities to evaluate complex relationships. Conceptually, ML has the benefit that it efficiently exploits complex and non-linear interactions between variables by minimizing the error between predicted and observed response variables and improve the accuracy of the models compared to standard approaches [15,16]. By using a large dataset available on practice assistants from our prior study, we aim to develop better understanding workplace factors, associated with chronic stress in practice assistants using machine learning. Thus, we compare four machine learning classifiers (random forest, support vector machine, K-nearest neighbors’, artificial neural network) with a standard logistic regression model using standard measurements to compare test accuracy, i.e. to derive the best prediction model for chronic stress in practice assistants in primary care. 

Regarding terminology, we like to point out that we use the term “prediction” as used in the context of machine learning: it refers to the output of an algorithm after it has been trained on a dataset and applied to new data to forecast the likelihood of a particular outcome. In contrast, in epidemiological analyses, a (risk) prediction model refers to a mathematical equation that uses patient characteristics (risk factors) to estimate the probability of a defined outcome prospectively.” 

b) Page 4, paragraph 1, lines 47-50: this information should go in the “Method” section.

Reply: Thank you for your advice. We now clarified that these results stem from our previous publication on chronic stress in GPs and practice assistants in the introduction. In addition, we included this information in the methods section.

Revised text: Please see page 7, lines 127-130 (document of revised Manuscript with track changes).

“We documented that 26.4% of the 550 practice assistants (PrAs) had high chronic stress, as well as 19.9% of the male (n = 141) and 35.6% of the female (n = 73) general practitioners (GPs) [9]. In this workforce, the average of workers with high chronic stress was 26.3% (n = 201).”

9. METHODS

Data source:

a) What categories are there among the "general practice personnel"? The authors must describe the sample used.

Reply: Thank you, we added the information in the introduction and methods section. 

Revised text: Please see Methods section, lines 119-127 (document of revised Manuscript with track changes): 

“The dataset used for the analyses was derived from our cross-sectional study addressing stress among general practice personnel (GPs, PrAs), which was performed among general practices belonging to the teaching practice network of the Institute for General Medicine, University Hospital Essen, Essen, Germany. A total of 764 professionals from 136 practices had taken part in the survey, which was performed in 2014. The design of the study and key results addressing the 214 GPs (practice owners and employed physicians) and 550 practice assistants (PrAs) (including medical secretaries and practice assistants in trainees) are published [9]. This analysis addresses chronic stress in 550 practice assistants (PrAs), which are the largest professional group in general practices.”

b) What was the target population? How was the sample chosen? The authors must specify it.

Reply: Thank you. We clarified this in the introduction and methods section. The target populations were 550 practice assistants (PrAs) from 136 teaching practice network.

Revised text: Please see page 7, lines 119-127 (document of revised Manuscript with track changes).

 “The dataset used for the analyses was derived from our cross-sectional study addressing stress among general practice personnel (GPs, PrAs), which was performed among general practices belonging to the teaching practice network of the Institute for General Medicine, University Hospital Essen, Essen, Germany. A total of 764 professionals from 136 practices had taken part in the survey, which was performed in 2014. The design of the study and key results addressing the 214 GPs (practice owners and employed physicians) and 550 practice assistants (PrAs) (including medical secretaries and practice assistants in trainees) are published [9]. This analysis addresses chronic stress in 550 practice assistants (PrAs), which are the largest professional group in general practices.”

10. DISCUSSION

a) Page 21, lines 272-297: The first two paragraphs do not contribute anything new and repeat information about the results.

Reply: We fully agree, thank you for pointing this out, we revised the text.

Revised text: Please see the pages 21-22, lines 311-354 (document of revised Manuscript with track changes).

“To the best of our knowledge, this study is the first to use machine learning for a better understanding of stress in primary care practice personnel. Comparing four common machine learning (ML) approaches to a classical statistical procedure, we showed that all four machine learning approaches provided more accurate models for the prediction of strain due to chronic stress than as standard regression analysis. Random forest showed the highest accuracy with workload, high demand to concentrate, and time pressure being the most important factors associated with chronic stress. These factors were also identified in other studies in the target populations GPs and GP practice personnel. Addressing job satisfaction, Harris et al. identified time pressure as the most frequent stressor in a study with 626 Australian practice staff in 96 general practices [12]. Studying 158 Canadian family physicians, Lee et al. determined the following occupational stressors as relevant: challenging patients, high workload, time limitations, competency issues, challenges of documentation and practice management and changing roles within the workplace [13,32]. Similarly, Hoffmann et al. showed that the work disruption was a negative relevant workplace factor in study with 550 practice assistants [33]. These stressors are described to influence poor physician well-being and adverse patient outcomes such as low patient satisfaction [34]. The relevance of such chronic psychological burden is tremendous as it was shown that physiological responses due to stress negatively affect e.g. memory, immune system functions, the function of the cardiovascular system, and brain electric activity [35,36].” 

11. Limitations

a) The information is from 2014. There was a global economic crisis that affected working conditions. It should be said since currently, working conditions are not equivalent.

Reply: The working conditions in German general practices did not change during the last years (except during the current pandemic). Workplaces are secure, there are no insolvencies of practices, and the income of practices is a mixture of reimbursement by the statutory health insurances and private patients. The migration influx in 2015 led to more patients in the system, but for each practice these were small numbers. Also, the gross national product remained stable for Germany (https://en.wikipedia.org/wiki/Gross_national_income). 

Revised text: Please see page 24, lines 399-401 (document of revised Manuscript with track changes).

“Although the data collection was conducted in 2014, the results still apply to German practices, except that the COVID pandemic likely increased workload and psychological burden, which we are currently evaluating in an ongoing study [11].”

12. REFERENCES

a) Many bibliographies are obsolete and some citations are incomplete. The bibliographic citations used are more than 5 years old (57,1%). The authors must update and arrange the bibliography.

Too many references do not meet the journal guidelines and that have errors. The authors should review this section.

Reply: Thank you, we reviewed the literature again and updated references. On the other hand, we continue to refer to important studies in the field even if they are older than 5 years. Now only 32.5% of the quotations are older than 5 years. 

Revised text: Please see section references.

---

## [Decision Letter · Decision Letter 1]

15 Apr 2021

Chronic stress in practice assistants: an Analytic approach comparing four machine learning classifiers with a standard logistic regression model

PONE-D-20-23593R1

Dear Dr. Bozorgmehr,

We’re pleased to inform you that your manuscript has been judged scientifically suitable for publication and will be formally accepted for publication once it meets all outstanding technical requirements.

Kind regards,

Alfredo Vellido

Academic Editor

PLOS ONE

Additional Editor Comments (optional):

Reviewers' comments:

Reviewer's Responses to Questions

**Comments to the Author**

1. If the authors have adequately addressed your comments raised in a previous round of review and you feel that this manuscript is now acceptable for publication, you may indicate that here to bypass the “Comments to the Author” section, enter your conflict of interest statement in the “Confidential to Editor” section, and submit your "Accept" recommendation.

Reviewer #2: All comments have been addressed

2. Is the manuscript technically sound, and do the data support the conclusions?

Reviewer #2: Yes

3. Has the statistical analysis been performed appropriately and rigorously? 

Reviewer #2: Yes

4. Have the authors made all data underlying the findings in their manuscript fully available?

Reviewer #2: Yes

5. Is the manuscript presented in an intelligible fashion and written in standard English?

Reviewer #2: Yes

6. Review Comments to the Author

Reviewer #2: Dear authors,

Thanks for your reply. The explanations of the authors are satisfactory. The paper has greatly improved its quality.

Congratulations on your work.

Best regards

7. PLOS authors have the option to publish the peer review history of their article (what does this mean?). If published, this will include your full peer review and any attached files.

Reviewer #2: No

---

## [Editor Report · Acceptance letter]

19 Apr 2021

PONE-D-20-23593R1 

Chronic stress in practice assistants: an Analytic approach comparing four machine learning classifiers with a standard logistic regression model 

Dear Dr. Bozorgmehr:

I'm pleased to inform you that your manuscript has been deemed suitable for publication in PLOS ONE. Congratulations! Your manuscript is now with our production department. 

Kind regards, 

on behalf of

Dr. Alfredo Vellido 

Academic Editor

PLOS ONE